# Evaluating Surface Properties and Cellular Responses to Surface-Treated Different Triple Periodic Minimal Surface L-PBF Ti6Al4V Lattices for Biomedical Devices

**DOI:** 10.3390/ijms26072960

**Published:** 2025-03-25

**Authors:** Viritpon Srimaneepong, Vorapat Trachoo, Suphalak Phothichailert, Supreda Suphanantachat Srithanyarat, Rangsini Mahanonda, Heil Norbert, Suppakrit Khrueaduangkham, Patcharapit Promoppatum, Thanaphum Osathanon

**Affiliations:** 1Department of Prosthodontics, Faculty of Dentistry, Chulalongkorn University, Bangkok 10330, Thailand; viritpon.s@chula.ac.th; 2Department of Oral and Maxillofacial Surgery, Faculty of Dentistry, Chulalongkorn University, Bangkok 10330, Thailand; vorapat.t@chula.ac.th; 3Center of Excellence for Dental Stem Cell Biology, Department of Anatomy, Faculty of Dentistry, Chulalongkorn University, Bangkok 10330, Thailand; joysuphalak07@gmail.com; 4Center of Excellence for Periodontology and Dental Implants, Department of Periodontology, Faculty of Dentistry, Chulalongkorn University, Bangkok 10330, Thailand; supreda.s@chula.ac.th; 5Immunology Research Center, Faculty of Dentistry, Chulalongkorn University, Bangkok 10330, Thailand; rangsini.m@chula.ac.th; 6Department of Periodontology, Faculty of Dentistry, Chulalongkorn University, Bangkok 10330, Thailand; 7DataPhysics Instruments GmbH, Raiffeisenstraße, 70794 Filderstadt, Germany; n.heil@dataphysics-instruments.com; 8Center for Lightweight Materials, Design, and Manufacturing, Department of Mechanical Engineering, Faculty of Engineering, King Mongkut’s University of Technology Thonburi (KMUTT), Bangmod, Bangkok 10140, Thailand; suppakrit.khrue@kmutt.ac.th

**Keywords:** cell behavior, inflammation, L-PBF, post-surface treatment, TPMS, Ti6Al4V

## Abstract

Triple periodic minimal surface lattices have been introduced to dental and medical devices. Numerous designs of these porous structures have been proposed, but the impact of the surface properties of the different topographic lattices are not fully understood. So, this study aimed to examine the cellular and inflammatory responses to different lattice designs, including strut-based and surface-based lattices. Human osteoblasts, human umbilical vein endothelial cells, and monocytes were used to evaluate cell proliferation, osteogenic differentiation, and inflammatory response on lattices after surface treatment strategies. Post-surface treatment of chemical etching, in addition to improving the surface roughness by removing some adhered metal powder, also modulated the surface energy. The lattice design had no significant impact on cell proliferation, but higher cell proliferation was found in post-surface treated lattices, regardless of topographic design. For angiogenesis, there was no difference in the release of pro-angiogenic growth factors between topographic designs or post-surface treatment groups. Moreover, lattices with the post-surface treatment were prone to have a lower inflammation phenotype when compared to an as-printed lattice, though not in a significant manner. This study implies that different topographic lattice designs may not have a major impact on bone ingrowth; nevertheless, post-surface treatment and surface properties of lattice may have an influence on a macrophage-induced inflammatory response.

## 1. Introduction

Autogenous bone graft is a well-known gold standard clinical modality for repairing bony defects. However, its success is dependent on the size of the bony defect(s) and potential donor site complications that can arise, such as donor site bleeding or morbidity. Metallic biomaterials are still one of the top options for manufacturing partial or total implants or prostheses not only to replace human joints such as the temporomandibular, hip, or knee joints but also to fill mandibular bone defects. Human bone is a complex living tissue with much lower stiffness than the common implant biomaterials, which can lead to a mismatch in elastic modulus between natural bone and biomaterials, especially the metallic ones. This mismatch may induce a stress shielding effect; hence, a strategy to reduce the stiffness of metallic implants is mandatory to mitigate this effect. There are alternative ways to modify the elastic modulus of metallic implants apart from the usual techniques to create bulk metal alloys. One of them is to alter the core microstructure during the manufacturing process. A porous microstructure in metallic implants has been developed during such a process, and it has been well accepted as a practical approach to tune the mechanical properties of these implants [1,2,3,4]. The porous metals can be prepared by many methods, such as powder metallurgy like foam casting or additive manufacturing. Laser powder bed fusion (L-PBF) is a prevalent additive manufacturing method utilized in the industry. The porous structure of metal has attracted interest from both dentistry and medical fields [5,6].

Titanium alloys have been the most extensively studied metallic biomaterials for medical or dental applications, but their inertness or high mechanical strength is still problematic. To overcome these challenges, the lattice-structured titanium alloys have been manufactured to produce porous titanium implants by additive manufacturing technologies such as laser powder bed fusion (L-PBF) or powder metallurgy (PM). Both have been well documented to lower the elastic modulus of bulk titanium alloys [2,3,4,6,7]. To improve the inertness of titanium alloy implants, the previous reports have shown that metallic porous structures can act as scaffolds to promote cell proliferation and bone ingrowth, as well as enhance vascular regeneration [8,9,10]. Chen et al. reported that porous metal with 60% porosity could induce better osteogenesis [8]. Additionally, Taniguchi et al. claimed that 600 μm pore size in porous titanium enhanced bone ingrowth and had sufficient compressive strength (42 MPa) for a load-bearing condition [11]. To further optimize the repair of bony defects, the structural topography of metallic implants can be modified or tuned according to the bone defect characteristics. The lattice structure can be classified into two types of lattices, surface-based and strut-based lattices. The triple periodic minimal surface (TPMS) lattice structure, such as the gyroid or I-graph wrapped package (IWP) lattice, represents a surface-based lattice due to its curvature. In contrast, the simple cubic lattice typifies the strut-based lattice. Porous titanium alloys with a pore size ranging from millimeter to micrometer have an impact on the proliferation of osteoblast or osteogenesis in animal studies [8,10,11,12]. Our preliminary studies evaluated the efficiency of cell proliferation on different TPMS lattices using computational simulations based on non-Newtonian computational fluid dynamics (CFD) [13]. Although different cellular responses were observed upon changes in lattice structure design and pore size, in silico cell behavior may not accurately reflect the characteristics of native bone cells. Hence, a comprehensive cell-based evaluation and optimization are required.

On the other hand, printed metallic surfaces can inevitably contain partially melted or unmelted metallic powder particles due to the laser powder bed fusion process. In certain instances, these unmelted powder particles can become loose and come off due to wear or corrosion, and this was found to trigger inflammation or macrophage-driven immunological responses [14,15]. This inflammatory process may interfere with cellular responses on lattice implants. Consequently, post-surface treatment of a printed metallic implant might be necessary to remove the unmelted powder particles from the printed surface. Thus, this study aimed to examine the biological responses of human osteoblasts, monocytes, and endothelial cells on different topographic designed lattice structures of Ti6Al4V ELI alloys, and the influence of post-surface treatment on cell behavior was also investigated.

## 2. Results

### 2.1. Surface Topography and Surface Energy

After post-surface treatment with acid etching, some of the partially melted or unmelted metal powder particles were removed from the printed lattice surface, resulting in a decreased surface roughness (Sa) for all lattice samples when compared to the non-etching lattices (Figure 1). Due to the geometric constraints of porous designs, laser scanning confocal microscopy was employed. It showed that surface roughness reduced after post-surface treatment. The surface roughness of the gyroid lattice decreased to a lesser extent than that of the other two lattice designs. This could be due to the surface morphology of the gyroid lattice that may influence the etching area and surface area measurement. While Table 1 presents the contact angle and surface energy of non-etching and etching groups. It suggested that the printed Ti6Al4V ELI alloy lattices displayed a slightly lower surface free energy after chemical etching compared to the non-etching samples, which were associated with a higher contact angle.

### 2.2. Cell Proliferation of Human Osteoblasts

Human osteoblasts were used to study the influence of lattice topography and post-surface treatment on cell growth. Figure 2 displays the three different topographies of titanium lattices with no discernible differences in cell proliferation of human osteoblasts observed during the 7 culture days. However, when porous lattices were surface treated with acid etchant, a different outcome was seen. After a longer time, the differences were evident. In the post-surface treatment group, the etched Ti6Al4V ELI alloy lattices promoted higher cell proliferation than non-etching porous structures, regardless of the type of design of each porous structure (*p* < 0.05). Although post-surface treatment could enhance cell proliferation, the topographic design appeared to have no effect on osteoblast growth.

### 2.3. Cell Attachment of Human Osteoblasts and HUVECs

From the MTT assay, even though a higher cell proliferation of human osteoblasts was observed in the titanium lattices with post-surface treatment, there were no obvious differences in cell attachment (Figure 3). Similar findings were also observed in HUVECs behavior. It was discovered that within the culture period of 3 h for human osteoblast and HUVECs, there was no significant difference in the relative percentage of cell attachment, regardless of lattice designs or post-surface treatment.

### 2.4. Gene Expression of HUVECs

Not only the new bone formation but also the development of new capillary blood vessels are essential for the osseointegration of implant biomaterials. Therefore, angiogenesis after 7 days was evaluated according to the expression of angiogenesis-related factors released by HUVECs (Figure 4). There were no meaningful differences in the mRNA expression of *VEGF*, *PDGF*, *CD34*, and *bFGF*, comparing between non-etching and etching groups and between lattice designs.

### 2.5. Osteogenic Differentiation Assay

Alkaline phosphatase activity and calcification assay (ARS staining).

Osteogenic differentiation was examined by ALP activity assessment and mineral deposition capacity (Figure 5). There were neither relevant differences in ALP activity between lattice designs nor between post-surface treatments (with/without chemical etching). To evaluate the late stage of osteogenic differentiation, Alizarin Red staining was performed to determine calcium deposition. There were no significant differences in calcium deposition between etching and non-etching groups. However, by calcification assay, the IWP-designed lattice appeared to favor higher mineralization than simple or gyroid lattices, regardless of post-surface treatment.

### 2.6. Inflammatory Response

The inflammatory response was determined by the transcriptome expression of immunomodulatory markers (*IDO*, *IFN-γ*, *IL-1β*, and *TNF-α*) in human monocyte (THP-1) cultures upon exposure to as-printed titanium lattices and post-treated titanium lattices (Figure 6). The mRNA expression was evaluated using real-time polymerase chain reaction at 24 h after seeding human monocyte cell line on the different designed lattices. The acid-etched titanium lattices noticeably suppressed the inflammatory transcriptome profile when compared to as-printed (non-etching) titanium lattices, as shown by the expression of *IL-1β* and *TNF-α*, especially in the gyroid lattice. IWP lattices exhibited slightly higher expression of *IL-1β* compared with the simple lattice in both non-etching and etching conditions, however, not significantly.

### 2.7. Scanning Electron Microscopy

Morphological features of human osteoblasts and HUVECs are shown in Figure 7 after 1 and 7 days of culture. In agreement with confocal micrographs, large numbers of adhered unmelted metallic powder particles were observed on the surface of non-etching (as-printed) lattice surfaces. Even though adhered unmelted particles could be removed by chemical etching post-surface treatment, a few unmelted powder particles with roughness remained on the lattice surface. In high magnification micrographs, both human osteoblasts and HUVECs appeared to be preferentially attached and grew between the waviness of powder particles. Both human osteoblasts and HUVECs could attach on the surface, as evidenced at day 1. The extension of lamellipodia and filopodia were observed in human osteoblasts and HUVECs on the porous samples. On day 7, both cells proliferated across the surfaces of the material. No sign of cellular process retraction or membrane rupture was observed. In addition, there was no marked difference in term of cell morphology, attachment, and spreading characters for both human osteoblasts and HUVECs on different lattice designs and surface treatments.

## 3. Discussion

Porous titanium has become an important biomaterial in the field of medical implant devices because of its unique physical and biological properties. Numerous studies have been conducted on porous materials, including the topographic designs that have been suggested [16,17,18]. The triple periodic minimal surface design of the porous structure is one of the surface-base lattice structure designs that have been incorporated into medical implants whose mechanical properties and biological response have been investigated [19,20,21]. There are many TPMS designs, and the impact of pore size and lattice design on cell behavior has been studied [11,12,21,22]. However, an optimal topographic design has not been clearly identified. Our previous in silico study [13] revealed that computational fluid dynamics (CFD) analysis could predict cell attachment and proliferation on different lattice designs. The proper pore size could facilitate fluid transport and lead to a potential increase in cell seeding. The surface topography and properties of metallic-implant-based biomaterials, including their surface chemistry in the present study, are not similar to those used in the CFD model; as a result, the understanding of their bio-functional roles in the presence of human bone tissue cells is necessary.

The present study investigated the effect of two distinct lattice structures−strut-based and surface-based lattice structures−on cell proliferation of human osteoblasts and HUVECs. It was discovered that while comparing the three designs, there was no influence of the topographic design of the lattice on cell growth. However, when the duration of the cell culture was up to 7 days, a higher cell growth could be observed on the porous structures post-surface-treated with chemical etching, regardless of the lattice design. This observation demonstrated that in comparison to the untreated Ti6Al4V ELI lattice, post-surface treatment could promote cell growth on the L-PBF lattice. It is well known that the surface quality of the L-PBF is affected by many processing parameters, such as build orientation, layer thickness, or speed [23]. During the printing process of L-PBF, the residual unmelted powder particles and partially melted powder particles inevitably adhere loosely to the L-PBF printed surface, as depicted in Figure 7. These unmelted metal particles will result in increased surface roughness. The surface roughness of L-PBF as-printed part is thought to be attributed to the ‘stair-case’ effect related to build orientation or particle size [24]. While the adherent unmelted powder is unavoidable, it can be improved by a number of methods such as mechanical polishing, chemical etching, electrochemical process, or even surface coating [25]. In this study, chemical etching was performed to address the structural restriction in the lattice samples, and this could partially remove unmelted powder particles, evidenced by a reduction in surface roughness to different extents of the designed lattice, observed using confocal microscopy (Figure 1). According to SEM analysis, both human osteoblasts and HUVECs exhibited a well-spread shape and drift into adherent powder particles (Figure 7). Therefore, the consequence of post-surface treatment with chemical etching could enhance more cell proliferation of human osteoblasts on the etched lattices, regardless of topographic design.

Next, to determine the correlation between the biophysical properties of lattice structures and cell response, the surface energy of both non-etching (as-printed) and etching lattices was investigated. Etched surfaces had lower surface energy (31.06 mN/m) than the non-etching surfaces (33.16 mN/m). This could be due to the reduced amount of unmelted powder particles facilitated by the chemical etching process, which, in turn, decreased the surface area and negatively affected the surface energy. This indicates that the post-surface treatment could enhance hydrophilic property, consequently influencing cell proliferation. While assessing the cell attachment of both human osteoblasts and HUVECs, no major differences could be observed either between lattice designs or between non-etching and etching lattices. Although a high surface energy of material is believed to be beneficial for cell adhesion, the present finding could imply that the level of cell attachment may not be affected only by the surface energy. Instead, it is also influenced by other factors such as surface topography, roughness, or cell type itself. Similar findings were also observed on the osteogenic differentiation in both the ALP activity and calcium deposition. To investigate the early stages of osteoblast differentiation and the late stages of mineralization, no differences were found. ALP activity and mineral deposition were comparable among the three different lattice designs, as well as between the post-surface treatment conditions. Both early and late phases of bone formation can be induced similarly irrespective of the topographic design or post-surface treatment with chemical etching.

Not only osteogenesis but also the angiogenesis mechanism is required for successful bone integration of orthopedic and dental implants. In this work, we examined the relationship between the surface chemistry of titanium lattices in terms of surface energy and the behaviors of cells by measuring the pro-angiogenic growth factors, including *VEGF*, *PDGF*, *CD34*, and *bFGF*, produced from HUVECs. The current data demonstrated that the expression of these angiogenic-related growth factors by HUVECs was not significantly impacted by the post-surface treatment and design of surface topography. In other words, the surface energy of the lattice surface might not be the main surface chemistry factor in the promotion of angiogenesis. In contrast to the previous study, it was claimed that high surface energy could induce more angiogenesis during osseointegration. It also showed that high surface energy with proper microrough surface topography could enhance the production of angiogenic growth factors [26]. The difference between their study and our current findings could be the result of the different surface modifications on titanium substrates rather than titanium lattices and different cell cultures, which were osteoblast-like cells. These could influence the response to synthesizing growth factors from different human cells.

As was already indicated, inevitably, the process of fabricating the titanium lattices utilizing L-PBF may result in the adherence of partially melted or unmelted metallic powder particles to printed surfaces as per SEM observations. These loosely attached powder particles or adhered unmelted powder could be affected by corrosion behavior. Therefore, if the unmelted powders loosen, these can trigger an immunological response and result in chronic inflammation [14,15,27]. This was among the primary reasons behind the post-surface treatment used in this investigation, which was to remove as many unmelted powder particles as possible. However, it is impossible to completely remove adhered partially melted or unmelted powder particles. Therefore, the human inflammatory response on the different design lattices with and without post-surface treatment was also evaluated. There were no major differences in the expression of pro-inflammatory cytokines between non-etching and etching groups, regardless of lattice design, but the printed lattice seemed to increase the cytokines within 24 h. In terms of inflammatory gene expression, both *IDO* and *IFN-γ* were upregulated, while inflammatory cytokines *IL-1β* and *TNF-α* were slightly increased. The low expression of inflammatory cytokines in the current investigation could be due to the biocompatibility property of titanium alloy powder itself as mentioned in the previous study [27]. It cannot, however, indicate that residual unmelted metallic powder is completely safe for human health.

An in vitro study has its own limitations and one of the limitations of this present study is related to lack of in vivo studies to validate the current findings. The residual partially melted or unmelted powder particles on lattice surfaces pose another drawback since they could affect the actual homogeneity of pore size for each lattice arrangement. This could have an impact on cellular response. This study failed to show a significant difference in cellular response across the various designed lattices; nevertheless, the post-surface treatment had an influence on the cell proliferation and inflammatory response. Our findings suggest that the surface chemistry may have greater impact on bone growth than lattice design. The optimization of surface treatment is more crucial than design lattice in improving cell behaviors and biocompatibility, providing valuable insights for the development of titanium-based biomedical devices.

## 4. Materials and Methods

### 4.1. Preparation of Titanium Lattices

Three different designs of titanium lattice disks sized 10 mm in diameter and 3 mm in thickness were designed using Materialise Mimics software (Materialise N.V. version 17.0.1, Leuven, Belgium). The three different structures include simple cubic (“Simple” will be represented here), representing the strut-based lattice, and two TPMS designed structures (IWP and gyroid lattices), representing surface-based lattice due to the structure curvature (Figure 8). All three designed lattices were fabricated with the same pore size of 600 μm and porosity of 67–70%. The geometrical characteristics of each lattice structure are shown in Table 2.

All lattice specimens were fabricated from Ti6Al4V ELI (Ti grade 23) alloy powder with 15–45 μm of particle size distribution (AP&C a GE Additive, Quebec, QC, Canada) using a laser powder bed fusion (L-PBF) machine (TruPrint 1000, Trumpf, Germany) with process parameters of laser power of 75 W, a scan velocity of 1000 mm/s, a hatch spacing of 110 µm, layer thickness of 20 µm, and laser spot diameter of 30 µm. After fabrication, all printed specimens were manually removed from the built platform by a wire-cut machine and cleaned inside an ultrasonic cleanser with deionized (DI) water to remove loose metal powder.

To perform a post-surface treatment on the L-PBF printed specimen surface, chemical etching by acid solution was performed in this study. According to previous studies [28,29], a hydrofluoric-containing solution was shown to be an effective etchant for titanium alloys. Hence, a nitric/hydrofluoric acid solution was used as the chemical etching agent in this study with the concentration of 4 M HF: 3.17 M HNO_3_ with a 1:2.5 wt% ratio following the study by Bezuidenhout et al. [28]. Each designed lattice group was divided into 2 subgroups—as-printed lattice specimen (no chemical etching) group and chemically etched lattice specimens as the post-surface group. For the etching process, the Ti6Al4V ELI lattice specimens were etched with the HF: HNO_3_ solution for 25 min under sonication at room temperature and then were later cleaned with ultrasonic cleanser using DI water. The Ti6Al4V ELI lattice specimens were sterilized through an autoclave process before the biological testing.

### 4.2. Analysis of Surface Chemistry

The surface topography of each lattice design, including acid-etched and non-etching lattice samples from all groups, was observed using laser scanning confocal microscopy (Lext OLS5000, Olympus, Tokyo, Japan). One lattice sample from each condition was randomly selected for observation to represent the surface topography of that condition. The areal surface measurement was performed to assess the surface roughness of the lattice samples using confocal microscopy.

To measure the surface energy of the L-PBF Ti6Al4V ELI alloy, solid Ti6Al4V ELI discs (without a porous structure) were fabricated using the same method and printing parameters. The solid samples were divided into 2 groups, non-etching and etching groups. The same protocol of post-surface treatment used for the lattice groups was followed. All samples were cleaned with DI water and dried before analysis. The contact angle measurements were made with an optical contact angle device (OCA 25, Dataphysics Instruments GmbH, Cologne, Germany). The dosing system for the liquids was an electronic single direct dosing system (SDDE), equipped with 3 electronic syringe dosing units (ESr-N). The ESr-N contained 3 syringes (Hamilton 500 µL) with the following liquids: diiodomethane, ethylene glycol, and thiodiglycol. The dispensed volume of each liquid was 0.5 µL.

All solid samples were analyzed by placing them on the sample table of the OCA25 and dispensing a drop of each liquid on one sample. The surface energy analysis was repeated for the other two samples. Using the software SCA (V6.2.26 Build 6026), the contour of the sessile drop was observed, and the contact angle was calculated using the ellipse fit function. A mean contact angle was calculated from the data of each liquid on the different samples. The mean contact angle value of each liquid was used to calculate the surface energy according to the OWRK approximation as the following equation.γ_total_ = γ_dispersive_ + γ_polar_(1)

γ_total_: the total surface energyγ_dispersive_: the dispersive (van der Waals) component of the surface energyγ_polar_: the polar component of the surface energy

### 4.3. Isolation and Cultivation of Human Osteoblasts

The study was approved by the Human Research Ethical Committee, Faculty of Dentistry, Chulalongkorn University (HREC-DCU 2023-013). The bone was collected by surgical removal according to the patient’s treatment plan and was obtained for cell isolation. The human osteoblasts were extracted using the tissue explant technique. The bone was dissected into small pieces and thoroughly washed with culture medium consisting of Dulbecco’s modified Eagle’s medium (DMEM) (Gibco, Billings, MT, USA) consisting of 10% fetal bovine serum (Gibco, Billings, MT, USA), 2 mM L-glutamine, and 100 unit/mL penicillin, 100 μg/mL streptomycin, and 250 μg/mL amphotericin B (Gibco, Billings, MT, USA) (growth medium). Cells were maintained in the growth medium, which was changed every 2 days, and incubated in a humidified atmosphere with 5% CO_2_ at 37 °C until confluence. Cells from passages 4–6 were used in the experiments. Lattice specimens were seeded with 2 × 10^4^ human osteoblasts/lattice.

### 4.4. Cell Culture of Human Umbilical Vein Endothelial Cells

Human umbilical vein endothelial cells (HUVECs) were kindly provided by Professor Rangsini Mahanonda, Department of Periodontology, Faculty of Dentistry, Chulalongkorn University, and cultured in complete endothelial cell growth medium-2 (EBM-2) supplemented with 1 × EGMTM-2 MV SingleQuotsTM Supplement Pack (CC-4147) (Lonza, Basel, Switzerland) and maintained at 37 °C in a 5% CO_2_ air humidified atmosphere. The culture medium was changed every 2 days until the cells reached around 95% confluence. The cells from passages 3–5 were used in all experiments.

### 4.5. Cell Culture of Human Monocyte Cells

THP-1 monocyte cells were cultured in RPMI 1640 (Thermo Fisher Scientific, Waltham, MA, USA) supplemented with 10% (*v*/*v*) FBS (Thermo Fisher Scientific), 1% (*v*/*v*) L-glutamine (Glutamax TM-1) (Thermo Scientific, USA), and 1% (*v*/*v*) antibiotic–antimycotic (Thermo Scientific). Culture medium was changed every 2 days and cells were incubated in a humidified atmosphere with 5% CO_2_ at 37 °C.

### 4.6. Proliferation and Cell Attachment Assays

Human osteoblasts and HUVECs were used to measure their attachment ability and proliferation on titanium scaffolds using an MTT ([3-(4, 5-dimethylthiazol-2-yl)-2, 5-diphenyltetrazolium bromide]) assay (Tocris Bioscience, Bristol, UK). Cells (2 × 10^4^ cells/0.5 mL) were seeded on sterile 3 scaffold designs and under 2 conditions (as-printed and chemically etched) in 48-well culture plates. The culture plates were incubated at 37 °C with 5% CO_2_ at the specified time intervals of 1, 3, and 7 days for the cell proliferation assay and 20 min, 1 h, and 3 h for cell attachment evaluation. After media removal, cells were incubated with 0.5 mg/mL MTT solution for 4 h at 37 °C. Afterwards, the insoluble formazan crystals were dissolved in a 0.5 mL dimethyl sulfoxide with glycine buffer. The absorbance at 570 nm was measured using a microplate reader (ELx800TM, BioTek, Washington, DC, USA).

### 4.7. Angiogenesis Assay

To investigate the angiogenesis of HUVECs on the different topography of Ti6Al4V ELI lattices. The HUVECs (2 × 10^5^ cells) were seeded on different designs and cultured with EBM-2 for 24 h. Total RNA was isolated using RiboExTM (GeneAll^®^, Seoul, Republic of Korea), and RNA was converted to cDNA using a reverse transcriptase enzyme kit (Promega, Madison, WI, USA). Afterward, qPCR was observed using FastStart SYBR Green Master kit (Roche Applied Science, Branford, CT, USA) with CFX Connect Real-Time PCR machine (Bio-Rad, Singapore). The angiogenesis growth factors were determined, including *VEGF*, *PDGF*, *CD34*, and *bFGF.* The expression values were normalized to *GAPDH* expression. Oligo primer sequences are shown in Table 3.

### 4.8. Osteogenic Differentiation Assay

#### 4.8.1. Alkaline Phosphatase (ALP) Activity Assay

Osteogenic differentiation on titanium scaffolds was assessed by measuring the ALP activity. Human osteoblasts (2 × 10^4^ cells/well) were seeded into a 48-well plate. Cells were cultured in an osteogenic induction medium containing 10% growth medium, 50 µg/mL ascorbic acid (Sigma-Aldrich, St. Louis, MO, USA), 5 mM beta-glycerophosphate (Sigma-Aldrich), and 100 nM dexamethasone (Sigma-Aldrich) for 7 days. The ALP activity was analyzed using an alkaline phosphatase assay kit (Abcam, Milpitas, CA, USA) according to the manufacturer’s protocol. Briefly, the cells were extracted using LIPA and mixed with 5 mM p-nitrophenyl phosphate (pNPP), ALP enzyme, stop solution for pNPP, and incubated at 25 °C for 60 min in the dark. The absorbance at 405 nm was measured by a microplate reader (ELx800TM, BioTek, Washington, DC, USA).

#### 4.8.2. Alizarin Red S

Calcium deposition on titanium scaffolds was examined using an Alizarin Red S staining assay. Cells were seeded (5 × 10^4^ cells/well) on sterile 3 patterned scaffolds under 2 different post-surface treatment conditions (non-etching and etching) in 48-well plates and cultured at 37 °C with 5% CO_2_ for 14 days. Cells matured under an osteogenic differentiation induction medium. At designated time points, the medium was removed, and cells were washed with DI water and subsequently stained with the Alizarin Red S for 5 min. Afterwards, the insoluble stained calcium was dissolved in 0.5 mL 10% cetylpyridinium chloride in 10 mM sodium phosphate. The absorbance at 570 nm was measured using a microplate reader (ELx800TM, BioTek, Washington, DC, USA).

### 4.9. Gene Expression of Inflammatory Markers

To investigate the inflammatory response of THP-1 monocytes on differently designed Ti6Al4V ELI lattices, the THP-1 cells were seeded on different designs in RPMI 1640 (5 × 10^4^ cells/well) for 24 h. The inflammatory cytokines (*IDO*, *IFN-γ*, *IL-1β*, and *TNF-α*) were measured using qPCR. The primer sequences are shown in Table 3.

### 4.10. Scanning Electron Microscopy

To observe cell morphology and attachment of human osteoblasts and HUVECs after seeding on different topographies of Ti6Al4V ELI lattices at day 1 and day 7. The samples were fixed with 3% glutaraldehyde in PBS for 30 min. Dehydration was performed using serial-graded ethanol (30–100%) and further added with hexamethyl disiloxane for 5 min and dried. The samples were observed using SEM (Quanta 250, FEI, Hillsboro, OR, USA).

### 4.11. Statistical Analysis

All experiments were repeated using cells derived from at least four different donors (*n* = 4). Kruskal–Wallis tests followed by a pairwise comparison or the analysis of variance (ANOVA) followed by multiple comparisons by Tukey test were used for statistical analysis depending on the normal distribution of the data. A *p*-value less than 0.05 indicates statistical significance except for surface roughness (Sa). The analysis was performed by using GraphPad Prism version 8 (GraphPad software, San Diego, CA, USA).

## 5. Conclusions

This investigation was performed on three different lattice designs fabricated by L-PBF. The effect of topographic design on the behaviors of human osteoblasts, HUVECs, and monocytes (THP-1) was investigated. The findings of the present study indicate the following:The surface chemistry of titanium lattice could be impacted by post-surface treatment, which influences cell differentiation and proliferation.The increase in cell proliferation was not affected by lattice design but could be enhanced by the post-surface treatment using chemical etching.Neither mineralization nor angiogenesis was influenced by lattice design and post-surface treatment.The L-PBF titanium lattice had a noticeable effect on the inflammatory response. The inflammatory response would also be influenced by the surface chemistry of printed titanium lattice.

## Figures and Tables

**Figure 1 ijms-26-02960-f001:**
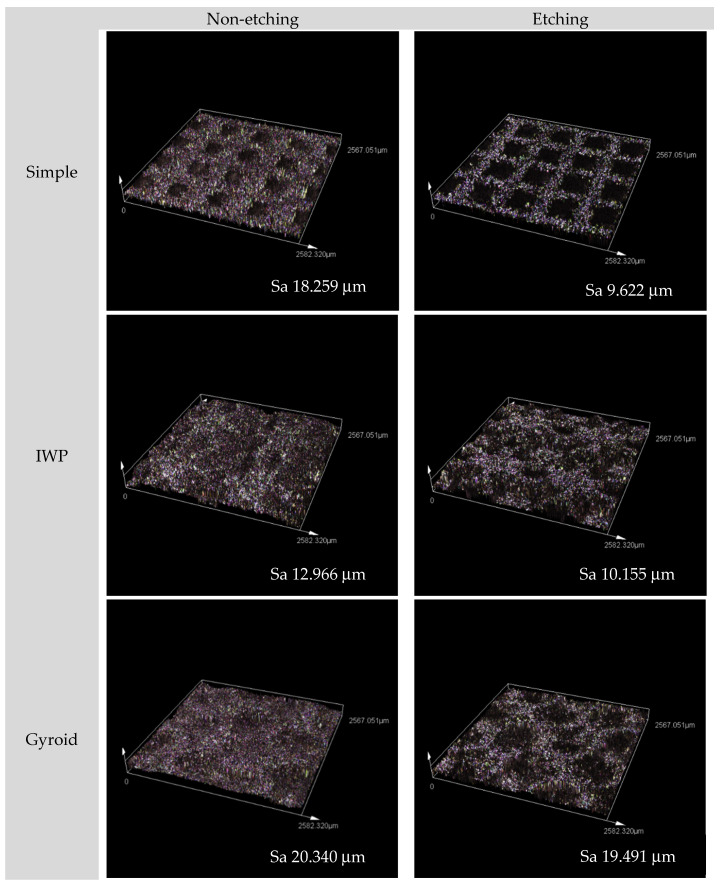
Surface topography micrographs of the three designed lattices (both non-etching and etching samples) observed by confocal microscopy with 20× magnification.

**Figure 2 ijms-26-02960-f002:**
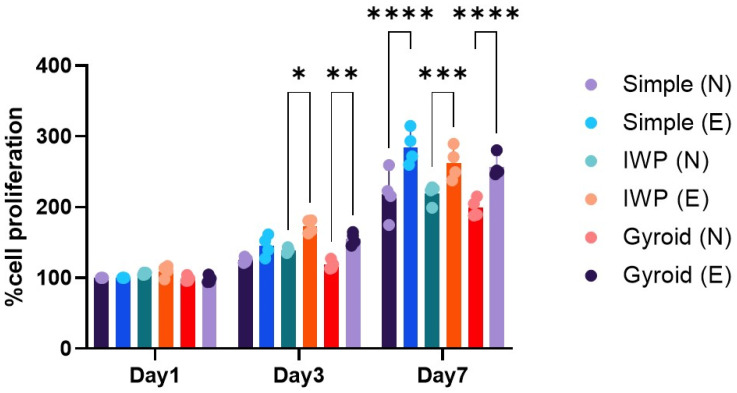
Cell proliferation of human osteoblasts at day 1, 3, and 7. (N: non-etching and E: etching) Statistically significant difference between the groups at * *p* < 0.05, ** *p* < 0.01, *** *p* < 0.001, **** *p* < 0.0001.

**Figure 3 ijms-26-02960-f003:**
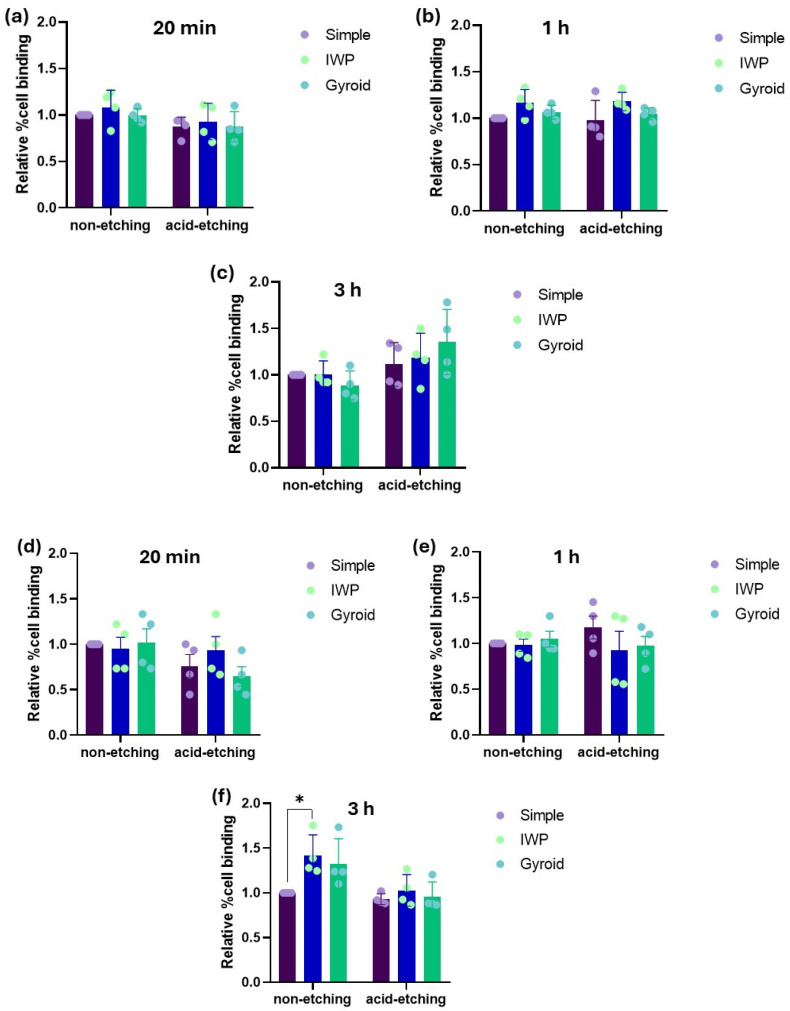
Relative percentage of cell attachment of human osteoblasts (**a**–**c**) and HUVECs (**d**–**f**) after 20 min, 1 h, and 3 h of culture. (* *p* < 0.05).

**Figure 4 ijms-26-02960-f004:**
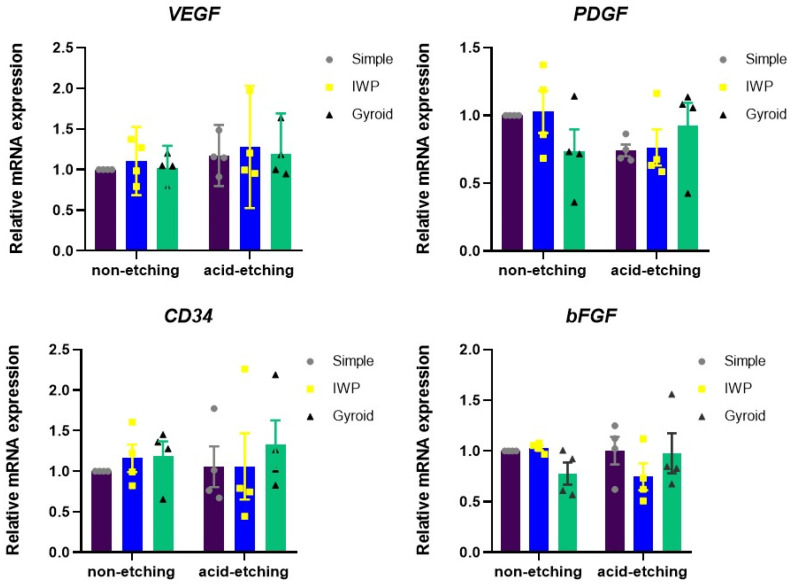
Expression of angiogenic growth factors assessed from HUVEC cultures after 24 h.

**Figure 5 ijms-26-02960-f005:**
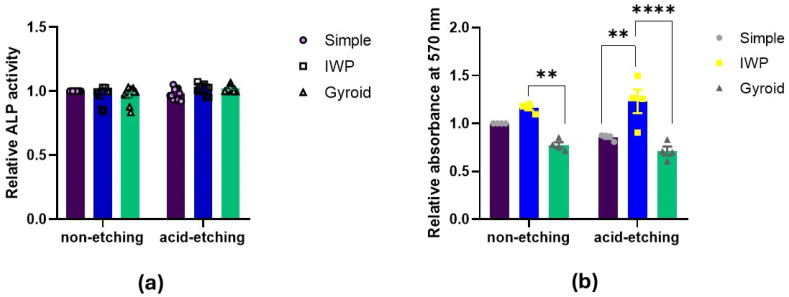
Osteogenic differentiation assessed by (**a**) ALP activity after 7 days and (**b**) calcification assay after 14 days (Statistically significant difference between the group at ** *p* < 0.01, **** *p* < 0.0001).

**Figure 6 ijms-26-02960-f006:**
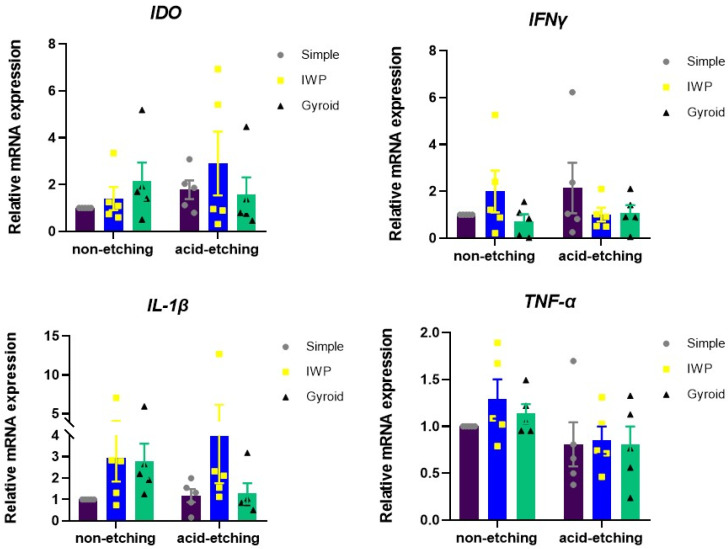
Inflammatory gene expression of THP-1 monocytes cultured on different lattice designs and in different conditions of post-surface treatment for 24 h. The total RNA isolation was performed. The mRNA expression of inflammatory related genes was examined using real-time polymerase chain reaction.

**Figure 7 ijms-26-02960-f007:**
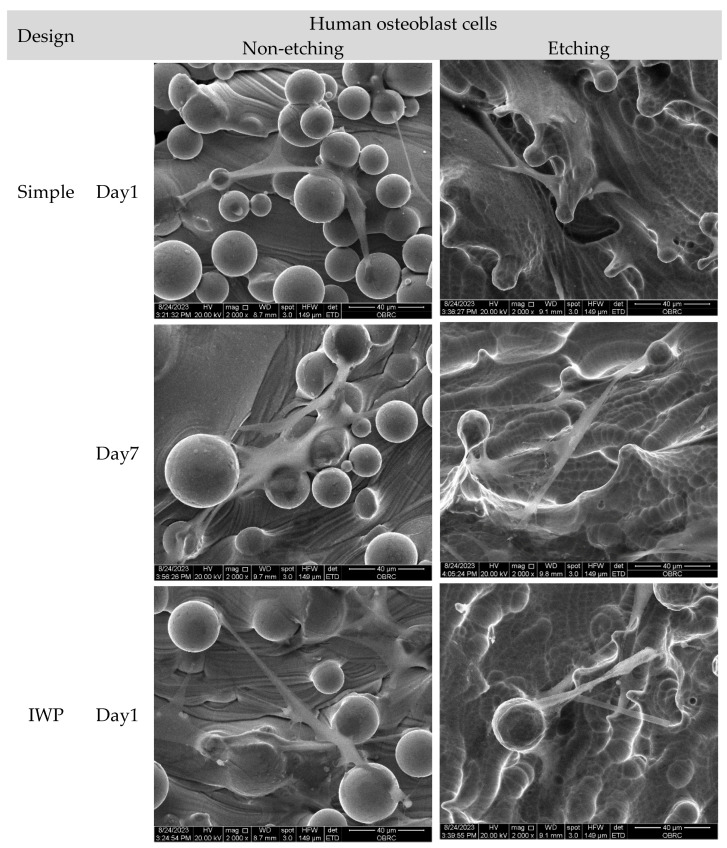
Scanning electron micrographs of human osteoblasts and HUVECs on non-etching (as-printed) and etching Ti6Al4V ELI alloy lattices of three different lattice designs (simple, IWP, and gyroid) after 1 and 7 days. (2000× magnification). Both cells successfully adhered to the surface on day 1, while cell spreading was seen on day 7. No significant difference was observed in cell morphology, attachment, and spreading characteristics of human osteoblasts and HUVECs across various designs and treatments.

**Figure 8 ijms-26-02960-f008:**
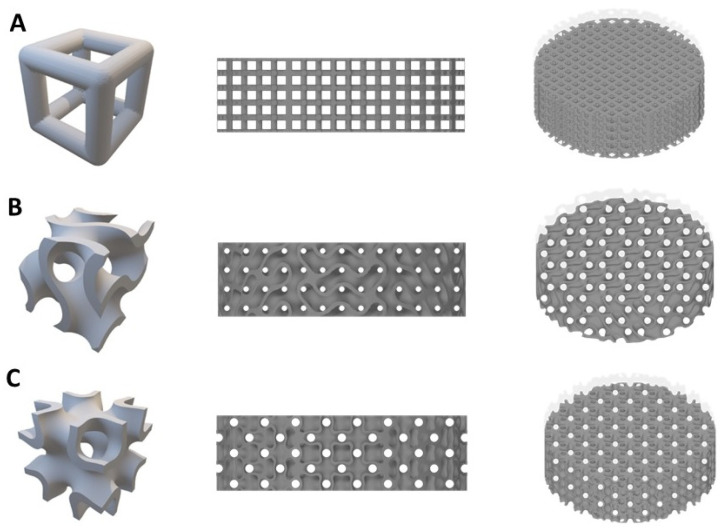
Models of the three lattice structures (**A**) simple cubic (**B**) gyroid (**C**) IWP.

**Table 1 ijms-26-02960-t001:** Contact angle and surface energies of the different post-surface treatments (non-etching and etching groups). Values are displayed as mean ± standard deviation for contact angle.

		Non-Etching 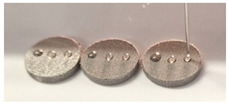	Etching 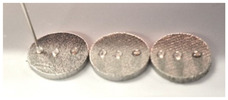
**Contact Angle [°]**	Diiodomethane	52.12 ± 0.81	55.52 ± 1.79
	Ethylene glycol	71.51 ± 3.63	73.65 ± 2.96
	Thiodiglycol	67.44 ± 2.18	71.01 ± 2.17
**Surface Energy (mN/m)**		33.16	31.03

**Table 2 ijms-26-02960-t002:** Geometrical parameters of Ti6Al4V ELI lattice structures.

Sample	Pore Size (µm)	Unit Cell Size (µm)	Wall Thickness (µm)	Surface Area (mm^2^)	Relative Density
Simple cubic	600	600	249	1050.73	0.33
Gyroid	600	1545	130	968.15	0.30
IWP	600	1228	100	1373.25	0.30

**Table 3 ijms-26-02960-t003:** Oligo primer sequences used for PCR.

Gene	Primer	Sequences
*bFGF*	Forward	5′ GGC TTC TTC CTG CGC ATC CAC 3′
	Reverse	5′ GGT AAC GGT TAG CAC ACA CTC CT 3′
*CD34*	Forward	5′ ACCACTAGCACTAGCCTTGC 3′
	Reverse	5′ CCTTCTTAAACTCCGCACAGC 3′
*GAPDH*	Forward	5′ TCATGGGTGTGAACCATGAGAA 3′
	Reverse	5′ GGCATGGACTGTGGTCATGAG 3′
*IDO*	Forward	5′ CATCTGCAAATCGTGACTAAG 3′
	Reverse	5′ GTTGGGTTACATTAACCTTCCTT 3′
*IFN-γ*	Forward	5′ CTA GGC AGC CAA CCT AAG CA 3′
	Reverse	5′ CAG GGT CAC CTG ACA CAT TC 3′
*IL1-β*	Forward	5′ TTCGAGGCACAAGGCACAA 3′
	Reverse	5′ CCATCATTTCACTGGCGAGC 3′
*PDGF*	Forward	5′ TCA GGT GGG TTA GAG ATG GAG T 3′
	Reverse	5′ GAA AGG AAC CAG AGG AAG AGG T 3′
*TNF-α*	Forward	5′ CACAGTGAAGTGCTGGCAAC 3′
	Reverse	5′ ACATTGGGTCCCCCAGGATA 3′
*VEGF*	Forward	5′ ATG AGG ACA CCG GCT CTG ACC A 3′
	Reverse	5′ AGG CTC CTG AAT CTT CCA GGC A 3′

## Data Availability

Data is contained within the article.

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
