# Peer review of "Evaluating Surface Properties and Cellular Responses to Surface-Treated Different Triple Periodic Minimal Surface L-PBF Ti6Al4V Lattices for Biomedical Devices"

_ijms, 2025, doi:10.3390/ijms26072960_

Round 1

Reviewer 1 Report

Comments and Suggestions for Authors

The present article present the influence of surface treatment and lattice geometry of 3D-printed metallic implants on the cell proliferation. The text reviews all the biological aspects of the subject, from the proliferation and attachment of the cells, to their differentiation and the inflammatory response associated with the material. The scientific presentation is sound and I only have minor remarks listed below.

Line 60: Foam casting is a process in powder metallurgy and not a separate method. Also, Reference 5 does not include additive manufacturing.
Line 62: Please provide appropriate reference.
Line 65: The paragraphs above explicit why the high mechanical strength is problematic but nothing is stated about the inertness. Please add an explanation on this point.
Line 79: Explicit the IWP abbreviation.
Line 109: Average roughness is sensitive to slanted samples. Please explicit the measures taken to avoid this pitfall. Also, consider adding a discussion about the significativity of the roughness differences.
Line 111: The surface energy is independent of the liquid. So "Regardless of the testing liquids" seems too much in this sentence.
Line 114: Consider adding a discussion about the significativity of the difference in surface energies.
Line 127: Explicit the N and E abbreviation in the caption.
Line 169: In my opinion, the difference of inflammatory response is not noticeable. It is true that in the gyroid lattice, the inflammation seems less pronounced by for the two other structures, ths samples display similar (or even higher) inflammation in the etched group compared to the non-etched one.
Lines 204-205: Two strange spiral symbols.
Line 216: Based on the SEM images, the unattached articles are around 20 to 40 µm large, which is in the same order as the layer size and the spot diameter. Consider adding a discussion about the contribution of unattached particles to the roughness.
Line 264: Only the bulk part of the pieces influences the mechanical properties. So, the powder particles attached should not have an impact on fatigue strength.
Line 279: I think that "in silico" should be "in vitro".
Line 296: There seems be be a problem with the size unit.
Line 307: Explicit the DI abbreviation.
Line 386: Missing exponent in "105".
Line 411: Please add the designated time points.
Line 432: Add the significant p-value for the surface roughness.

Comments on the Quality of English Language

Should "Post-surface treatment" rather be "Post-processin surface treatment"? Please check and replace in the whole text if necessary.

Line 305: Missing "of" before "30 µm".
Line 315: "chemical etched lattice specimens" shoud be "chemically etched lattice specimens.
Line 403: The "a" at the end of the line seems not wanted. Please check.
Line 409: The syntax of "Cells were seeded on sterile 3 patterned scaffolds under 2 etching conditions" seems odd. Please check.

Author Response

Line 60: Foam casting is a process in powder metallurgy and not a separate method. Also, Reference 5 does not include additive manufacturing.

Response:  Thank you for pointing this out.  We have rewritten. (Line 59-60)

Line 62: Please provide appropriate reference.

Response:   Thank you for your valuable suggestion.  We have rewritten in the introduction section including references (5, 6). (Line 61-63)

References

  1. S. Bencharit, W.C Byrd, S Altarawneh, B Hosseini, A Leong, G Reside, T Morelli. and S Offenbacher. (2014), Porous Tantalum Trabecular Metal Dental Implants. Clin Implant Dent Relat Res 16 (2014) 817-826.
  2. L.E. Murr, S.M. Gaytan, F. Medina, H. Lopez, E. Martinez, B.I. Machado, D.H. Hernandez, L. Martinez, M.I. Lopez, R.B. Wicker, J. Bracke, Next-generation biomedical implants using additive manufacturing of complex, cellular and functional mesh arrays, Philos Trans A Math Phys Eng Sci 368(1917) (2010) 1999-2032.

Line 65: The paragraphs above explicit why the high mechanical strength is problematic but nothing is stated about the inertness. Please add an explanation on this point.

Response:   Thank you for your valuable suggestion.  We have added more explanations. (Line 69-72) as follows

“To improve the inertness of titanium alloy implants, the previous reports have shown that metallic porous structures can act as scaffolds to promote cell proliferation and bone ingrowth as well as enhance vascular regeneration [8-10].”

Line 79: Explicit the IWP abbreviation.

Response:   Thank you for your suggestion. We have added full term of IWP as “I-graph Wrapped Package” (Line 79)

Line 109: Average roughness is sensitive to slanted samples. Please explicit the measures taken to avoid this pitfall. Also, consider adding a discussion about the significance of the roughness differences.

Response:  Thank you for pointing this out.  We have explained the measurement of surface roughness in Materials and Methods section (Line 344-349).  We have rewritten in the result section (Line 111-116) and discussed the significance of different surface roughness in the discussion section (Line 229-241).

Line 111: The surface energy is independent of the liquid. So "Regardless of the testing liquids" seems too much in this sentence.

Response:   Thank you for pointing this out. We agree with this comment. Therefore, we have removed it and rewritten the sentences.  (Line 117-119)

Line 114: Consider adding a discussion about the significance of the difference in surface energies.

Response:   Thank you for your valuable comment.  We have added more discussion about significant differences of surface energy in the discussion part. (Line 251-253)

Line 127: Explicit the N and E abbreviations in the caption.

Response:   Thank you for pointing this out.  We have added full terms of N and E in the caption of Figure 2 . (Line 132-133)

Line 169: In my opinion, the difference of inflammatory response is not noticeable. It is true that in the gyroid lattice, the inflammation seems less pronounced by for the two other structures, the samples display similar (or even higher) inflammation in the etched group compared to the non-etched one.

Response:   Thank you for your valuable comment. We have rewritten the results in the inflammatory section as follows (Line 173-182).

“The inflammatory response was determined by the transcriptome expression of immunomodulatory markers (IDO, IFN-g, IL-1β, and TNF-a) in human monocyte (THP-1) cultures upon exposure to as-printed titanium lattices and post-treated titanium lattices (Fig. 7). The mRNA expression was evaluated using real-time polymerase chain reaction at 24 h after seeding human monocyte cell line on the different designed lattices. The acid-etched titanium lattices noticeably suppressed the inflammatory transcriptome profile when compared to as-printed (non-etching) titanium lattices, as shown by the expression of IL-1β and TNF-a, especially in Gyroid lattice. IWP lattices exhibited slightly higher expression of IL-1β compared with Simple lattice in both non-etched and etched conditions, however, not significantly.”

Lines 204-205: Two strange spiral symbols.

Response:   Sorry for this error. We have corrected them. (Line 222-223)

Line 216: Based on the SEM images, the unattached particles are around 20 to 40 µm large, which is in the same order as the layer size and the spot diameter. Consider adding a discussion about the contribution of unattached particles to the roughness.

Response:  Thank you for pointing this out.  We have added more discussion about the unmelted particles related to surface roughness.  (Line 229-241) as follows

“It is well known that the surface quality of the L-PBF is affected by many processing parameters, such as build orientation, layer thickness, or speed [25].  During the printing process of L-PBF, the residual unmelted powder particles and partially melted powder particles are inevitably adhere loosely to the L-PBF printed surface, as depicted in Figure 7. These unmelted metal particles will result in increased surface roughness. As the surface roughness of L-PBF as-printed part is thought to be attributed to the ‘stair-case’ effect related to build orientation, or particle size [26]. While the adherent unmelted powder is unavoidable, it can be improved by a number of methods such as mechanical polishing, chemical etching, electrochemical process, or even surface coating [27]. In this study, chemical etching was performed to address the structural restriction in the lattice samples, and this could partially remove unmelted powder particles, evidenced by a reduction in surface roughness observed using laser scanning confocal microscopy (Figure 1).”

Line 264: Only the bulk part of the pieces influences the mechanical properties. So, the powder particles attached should not have an impact on fatigue strength.

Response:   Thank you for pointing this out. We agree with this comment. Therefore, we have rewritten in the discussion part.  (Line 283-286)

Line 279: I think that "in silico" should be "in vitro".

Response:   We agree.  We have changed it to be “in vitro”.  (Line 299)

Line 296: There seems to be a problem with the size unit.

Response:   Sorry for this mistake.  We have corrected it.  (Line 318)

Line 307: Explicit the DI abbreviation.

Response:   We have added full term of DI. (Line 329)

Line 386: Missing exponent in "105".

Response:   Sorry for this mistake.  We have corrected it to be 105.  (Line 411)

Line 411: Please add the designated time points.

Response:   We have added the designated time points in the Materials and Methods section. (Line 421)  Alkaline phosphatase (ALP) activity assay was investigated at 7 days while  Alizarin Red S was observed at 14 days.

Line 432: Add the significant p-value for the surface roughness.

Response:   Thank you for pointing this out. This study did not aim to compare the surface roughness of various designed lattice samples. The surface roughness was presented as a descriptive result before and after post-surface treatment, therefore, a statistical p-value for surface roughness is not applicable.

Response to Comments on the Quality of English Language

Should "Post-surface treatment" rather be "Post-processing surface treatment"? Please check and replace in the whole text if necessary.

Response:  Agree. We have replaced all “Post-processing” with “Post-surface” in the whole text.

Line 305: Missing "of" before "30 µm".

Response:  Sorry for this mistake.  We have added it. (Line 327)

Line 315: "chemical etched lattice specimens" should be "chemically etched lattice specimens.

Response:  We agree and have corrected it. (Line 337)

Line 403: The "a" at the end of the line seems not wanted. Please check.

Response:  We agree and have removed it. (Line 428)

Line 409: The syntax of "Cells were seeded on sterile 3 patterned scaffolds under 2 etching conditions" seems odd. Please check.

Response:   Sorry for the unclear sentence.  We have rewritten this sentence. (Line 434-436) as follows

“Cells were seeded (5x104 cells/well) on sterile 3 patterned scaffolds under 2 different post-surface treatment conditions (non-etching and etching) in 48-well plates and cultured at 37ºC with 5% CO2 for 14 days.”

Reviewer 2 Report

Comments and Suggestions for Authors

The paper submitted by Viritpon Srimaneepong et al. aims “to examine the cell and inflammatory responses to different lattice designs” for the production of medical devices. In particular, different topographic designed lattice structures based on the Ti6Al4V ELI alloy have been produced and the effects of a post-processing treatment (etching) was investigated.

Some aspects deserve further explanation.

  1. From a general point of view, the meaning of the term "biomechanical" in relation to "response" in the title is not clear. Apart from other surface characterization techniques, no biomechanical test has been described in the paper.
  2. The authors should better address the relevance of their work; taken together, the results do not show significant differences between the examined surfaces. Therefore, it is not clear if (and how) their work can be useful in order to improve the quality of Ti-based biomedical devices.
  3. Based on the experimental results, the final sentence in the abstract appears somewhat emphatic: greater caution is suggested.
  4. In Figure 1, the average roughness (Sa) is mentioned but the authors did not explain how it was measured (Materials and Methods section); moreover, Sa is not enough to exhaustively assess surface roughness.
  5. In Figure 2, abbreviations and asterisks need to be explained: the figure and the corresponding caption have to be self-explaining.
  6. Figure 3: the choice for the experimental time-points has to be justified: to my knowledge, 20 minutes is not sufficient to observe the effects of cell adhesion.
  7. Figure 7 requires a deeper explanation, in both the caption and the text. The authors have to specify what is depicted in each sketch, to allow the reader understand similarities/differences in the behavior of the cells onto the investigated surfaces at different time-points.
  8. Materials and Methods: the temperature for the etching treatment has to be specified.

For the above-mentioned reasons, major revisions are suggested.

Comments on the Quality of English Language

The quality of the written English could be improved.

Author Response

1. From a general point of view, the meaning of the term "biomechanical" in relation to "response" in the title is not clear. Apart from other surface characterization techniques, no biomechanical test has been described in the paper.

Response:  Thank you for pointing this out.  We agree and have modified the title as follows

Evaluating Surface Properties and Cellular Responses to Surface-Treated Different Triple Periodic Minimal Surface L-PBF Ti6Al4V Lattices for Biomedical Devices

2. The authors should better address the relevance of their work; taken together, the results do not show significant differences between the examined surfaces. Therefore, it is not clear if (and how) their work can be useful in order to improve the quality of Ti-based biomedical devices.

Response:  Thank you for your valuable comments. We agree with this comment and have rewritten in the discussion part. (Line 303-308) as follows

“This study could not demonstrate a significant difference on cellular response across the various designed lattices, yet it is impossible to assert definitely that the lattice topography had no substantial impact on enhanced bone development. Our findings could imply that surface properties play a significant role in improved cell proliferation. Consequently, these could be advantages for design and improvement of the titanium-based biomedical devices.”

3. Based on the experimental results, the final sentence in the abstract appears somewhat emphatic: greater caution is suggested.

Response:  Thank you for pointing this out. We agree with this comment. Therefore, we have rewritten in the abstract. (Line 39-41) as follows

“This study implies that different topographic lattice designs may not have a major impact on bone ingrowth, nevertheless, post-surface treatment and surface properties of lattice may have an influence on a macrophage-induced inflammatory response.”

4. In Figure 1, the average roughness (Sa) is mentioned but the authors did not explain how it was measured (Materials and Methods section); moreover, Sa is not enough to exhaustively assess surface roughness.

Response:  

Thank you for pointing this out and we agree.  We have rewritten in the Materials and Methods section (Analysis of Surface Chemistry) to explain how to measure the surface roughness. (Line 344-349) 

Due to the porous structure of lattice samples, Laser Scanning Confocal Microscope was employed to examine the surface topography and quantify the surface roughness, as previously documented (1).  However, this study aimed solely to descriptively present the surface roughness, rather than for comparative purposes.  The Sa values were not subjected to statistical analysis.

Reference

1.Lange, D.A., Jennings, H.M. & Shah, S.P. Analysis of surface roughness using confocal microscopy. Journal of Materials Science 28, 3879–3884 (1993).

5. In Figure 2, abbreviations and asterisks need to be explained: the figure and the corresponding caption have to be self-explaining.

Response:  Thank you for pointing this out. We have explained the abbreviation and asterisk in the caption of Figure 2. (Line 132-133)

6. Figure 3: the choice for the experimental time-points has to be justified: to my knowledge, 20 minutes is not sufficient to observe the effects of cell adhesion.

Response:  The choice of a 20-minute time point was based on previous studies demonstrating early adhesion events within this timeframe. Our focus was on capturing the initial interactions between cells and the substrate. Additionally, preliminary experiments indicated measurable adhesion changes within 20 minutes. However, we acknowledge that longer time points might provide further insights and are open to discussing this aspect in more detail. The 20 minutes-time points for cell adhesion evaluation were previously reported in several publications (1-3)

References

1.Osathanon T, Sawangmake C, Ruangchainicom N, Wutikornwipak P, Kantukiti P, Nowwarote N, Pavasant P. Surface properties and early murine pre-osteoblastic cell responses of phosphoric acid modified titanium surface. J Oral Biol Craniofac Res. 2016 Jan-Apr;6(1):2-9. doi: 10.1016/j.jobcr.2015.12.005. Epub 2015 Dec 24.

2.Osathanon T, Bespinyowong K, Arksornnukit M, Takahashi H, Pavasant P.
Human osteoblast-like cell spreading and proliferation on Ti-6Al-7Nb surfaces of varying roughness. J Oral Sci. 2011 Mar;53(1):23-30. 

3.Osathanon T, Bespinyowong K, Arksornnukit M, Takahashi H, Pavasant P. Ti-6Al-7Nb promotes cell spreading and fibronectin and osteopontin synthesis in osteoblast-like cells. J Mater Sci Mater Med. 2006 Jul;17(7):619-25. 

7. Figure 7 requires a deeper explanation, in both the caption and the text. The authors have to specify what is depicted in each sketch, to allow the reader understand similarities/differences in the behavior of the cells onto the investigated surfaces at different time-points.

Response:  Thank you very much for your suggestion. We have written more explanation on the behavior of cells in the result part and also the caption of Figure 7 as follow (Line 191-197)

“Both human osteoblasts and HUVECs could attach on the surface as evidenced at day 1. The extension of lamellipodia and filopodia were observed in human osteoblasts and HUVECs on the porous samples. On day 7, both cells proliferated across the surfaces of material. No sign of cellular process retraction or membrane rupture was observed. In addition, there was no marked difference in term of cell morphology, attachment, and spreading characters for both human osteoblasts and HUVECs on different lattice designs and surface treatments.”

For the caption of Figure 7 (Line 200-205)

Figure 7. Scanning electron micrographs of human osteoblasts and HUVECs on non-etching (as-printed) and etching Ti6Al4V ELI alloy lattices of three different lattice designs (Simple, IWP, and Gyroid) after 1 and 7 days. (2,000x magnification). Both cells successfully adhered to the surface on day 1 while cell spreading was seen on day 7. No significant difference was observed in cell morphology, attachment, and spreading characteristics of human osteoblasts and HUVECs across various designs and treatments.

8. Materials and Methods: the temperature for the etching treatment has to be specified.

Response:  Thank you for pointing this out. We agree with this comment. Therefore, we have included the temperature of etching condition in the Materials and Methods part. (Line 338-340)

“For the etching process, the Ti6Al4V ELI lattice specimens were etched with the HF:HNO3 solution for 25 minutes under sonication at room temperature and then were later cleaned with ultrasonic cleanser using DI water.”

Round 2

Reviewer 2 Report

Comments and Suggestions for Authors

The paper submitted by Viritpon Srimaneepong et al. has been amended following the reviewer’s suggestions. However, some aspects of the paper need further revisions. In particular, the authors have not been able to explain the usefulness of their findings. Actually, it is already well acknowledged that “surface properties play a significant role in improved cell proliferation” (line 306), but they did not observe different cell behaviors in contact with differently treated surfaces (“This study could not demonstrate a significant difference on cellular response across the various designed lattices”, line 303). The sentence “it is impossible to assert definitely that the lattice topography had no substantial impact on enhanced bone development” (line 304) is inappropriate in a scientific paper.

Minor revisions are necessary before considering the paper for publication.

Comments on the Quality of English Language

The quality of the written English is acceptable

Author Response

Comment:  The paper submitted by Viritpon Srimaneepong et al. has been amended following the reviewer’s suggestions. However, some aspects of the paper need further revisions. In particular, the authors have not been able to explain the usefulness of their findings. Actually, it is already well acknowledged that “surface properties play a significant role in improved cell proliferation” (line 306), but they did not observe different cell behaviors in contact with differently treated surfaces (“This study could not demonstrate a significant difference on cellular response across the various designed lattices”, line 303). The sentence “it is impossible to assert definitely that the lattice topography had no substantial impact on enhanced bone development” (line 304) is inappropriate in a scientific paper.

Response:   Thank you for your valuable comments.  We have rewritten following your comments as follows (Line 229-309)

            An in vitro study has its own limitations and one of the limitations of this present study is related to lack of in vivo studies to validate the current findings. The residual partially melted or unmelted powder particles on lattice surfaces pose another drawback since they could affect the actual homogeneity of pore size for each lattice arrangement. This could have an impact on cellular response. This study failed to show a significant difference in cellular response across the various designed lattices, nevertheless, the post-surface treatment had an influence on the cell proliferation and inflammatory response. Our findings suggest that the surface chemistry may have greater impact on bone growth than lattice design. The optimizing surface treatment is more crucial than design lattice in improving cell behaviors and biocompatibility, providing valuable insights for the development of titanium-based biomedical devices.